# SAFENet: A Secure, Accurate and Fast Neural Network Inference

Qian Lou[1], Yilin Shen[2], Hongxia Jin[2], and Lei Jiang[1]

[1]{louqian, jiang60}@iu.edu
[2]{yilin.shen, hongxia.jin}@samsung.com
[1]Indiana University Bloomington, [2]Samsung Research America

## ABSTRACT

The advances in neural networks have driven many companies to provide prediction services to users in a wide range of applications. However, current prediction systems raise privacy concerns regarding the user's private data. A cryptographic neural network inference service is an efficient way to allow two parties to execute neural network inference without revealing either party's data or model. Nevertheless, existing cryptographic neural network inference services suffer from enormous running latency; in particular, the latency of communication-expensive cryptographic activation function is 3 orders of magnitude higher than plaintext-domain activation function. And activations are the necessary components of the modern neural networks. Therefore, slow cryptographic activation has become the primary obstacle of efficient cryptographic inference. In this paper, we propose a new technique, called SAFENet, to enable a Secure, Accurate and Fast nEural Network inference service. To speedup secure inference and guarantee inference accuracy, SAFENet includes channel-wise activation approximation with multiple-degree options. This is implemented by keeping the most useful activation channels and replacing the remaining, less useful, channels with various-degree polynomials. SAFENet also supports mixed-precision activation approximation by automatically assigning different replacement ratios to various layer; further increasing the approximation ratio and reducing inference latency. Our experimental results show SAFENet obtains the state-of-the-art inference latency and performance, reducing latency by $38\% \sim 61\%$ or improving accuracy by $1.8\% \sim 4\%$ over prior techniques on various encrypted datasets.

## 1 INTRODUCTION

Neural network inference as a service (NNaaS) is an effective method for users to acquire various intelligent services from powerful servers. NNaaS includes many emerging, intelligent, client-server applications such as smart speakers, voice assistants, and image classifications Mishra et al. (2020). However, to complete the intelligent service, the clients need to upload their raw data to the model holders. The network model holders in the server are able to access, process users' confidential data from the clients, and acquire the raw inference results, which potentially violates the privacy of clients. So there is an urgent requirement to ensure the confidentiality of users' financial records, healthy-care data and other sensitive information during NNaaS.

Modern cryptography such as Homomorphic Encryption (HE) by Gentry et al. (2009) and Multi-Party Computation (MPC) by Yao (1982) enables secure inference services that protect the user's private data. During secure inference services, the provider's model is not released to any users and the user's private data is encrypted by HE or MPC. CryptoNets proposed by Gilad-Bachrach et al. (2016) is the first HE-based secure neural network on encrypted data; however, its practicality is limited by enormous computational overhead. For example, CryptoNets takes $\sim 298$ seconds to perform one secure MNIST image inference on a powerful server; its latency is 6 orders of magnitude longer than the unencrypted inference. MiniONN by Liu et al. (2017) and Gazelle by Juvekar et al. (2018) prove that using a hybrid of HE and MPC it is possible to design a low-latency, secure inference. Although Gazelle significantly reduces the MNIST inference latency of

CryptoNets into $\sim 0.3$ seconds, it is still far from practical on larger dataset such as CIFAR-10 and CIFAR-100, due to heavy HE encryption protocol and expensive operations. For instance, Gazelle requires $\sim 240$ seconds latency and $\sim 8.3$ GB communication to perform ResNet-32 on the CIFAR-100 dataset. NASS by Bian et al. (2020), CONAD by Shafran et al. (2019) and CryptoNAS by Ghodsi et al. (2020) are proposed to design cryptography-friendly neural network architectures, but they still suffer from heavy encryption protocol in the online phase. Delphi by Mishra et al. (2020) significantly reduces the online latency by moving most heavy cryptography computations into the offline phase. Offline computations can be pre-processed in advance.

The State-of-the-art cryptographic inference service Delphi by Mishra et al. (2020) still suffers from enormous online latency; this is because a big communication overhead between the user and the service provider is required to support cryptographic $ReLU$ activations. Our experiments show that the communication overhead is proportional to $ReLU$ units in the whole neural network. Delphi attempts to reduce inference latency by replacing expensive $ReLU$ with cheap polynomial approximation. Unfortunately, most $ReLU$ units are found to be difficult to substitute without incurring a loss of accuracy. The accuracy will be dramatically decreased as more $ReLU$ units are approximated by polynomials. Specifically, Delphi only replaces $\sim 42\%\ ReLU$ numbers on a CNN-7 network (detailed in Section 6.2 of MiniONN by Liu et al. (2017)) and $\sim 20\%\ ReLU$ numbers on ResNet-32 network, with $< 1\%$ accuracy decrease. When Delphi approximates more $ReLU$ units, $> 3\%$ inference accuracy will be lost compared to an all-$ReLU$ model. If accuracy loss is constrained, non-linear layers still occupy almost 62% to 74% total latency in CNN-7 and ResNet-32 networks. Therefore, slow, non-linear layers are still the obstacle of a fast and accurate secure inference.

**Our contribution.** One key observation is that the layer-wise activation approximation strategy in Delphi is too coarse-grained to replace the *bottleneck layers* in which the $ReLU$ units are mainly located, e.g. the first layer in CNN-7 occupies $> 58\%\ ReLU$ units. The channels in *bottleneck layers* are difficult to completely replace without a small accuracy loss. To meet accuracy constraints and speedup secure inference, SAFENet includes a more fine-grained channel-wise activation approximation to keep the most useful activation channels within each layer and replace the remaining, less important, activation channels by polynomials. In this way, only partial channels in each layer will be approximated, which is approximate-friendly for *bottleneck layers*. Another contribution of SAFENet is that automatic multiple-degree polynomial exploration in each layer is supported, compared to prior works using only degree-2 polynomials. Additionally, SAFENet enables mixed-precision activation approximation by assigning different approximation ratios to various layers, which further replaces more $ReLU$ units with cheap polynomials. Our results show that under the same accuracy constraints, SAFENet obtains state-of-the-art inference latency, reducing latency by $38\% \sim 61\%$, or improving accuracy by $1.8\% \sim 4\%$ over the prior techniques.

## 2 BACKGROUND AND RELATED WORK

**Threat Model and Cryptographic Primitives.** Our threat model is the same as previous work Delphi by Mishra et al. (2020). More specifically, we consider the service holder as a semi-honest cloud which attempts to infer clients input information but follows the protocol. The server holds the Convolutional Neural Network (CNN) model and the client holds the input to the network. For linear computations, the client encrypts input and sends it to the server using a HE scheme by Mishra et al. (2020), and then the server returns encrypted output to the client. The client decrypts and decodes the received output. The secret sharing (SS) in Delphi is used to protect the privacy of intermediate results in the hidden layers. Then garbled circuits (GC) guarantees the data privacy in the activation layers, and SS is used to securely combine HE and GC. Other than GC, Beaver's multiplicative Triples (BT) proposed by Beaver (1995) is used to implement approximated activation using secure polynomials. BT-based polynomial approximation for $ReLU$ is 3-orders of magnitude cheaper than GC-based $ReLU$ units on average, so it is used to design approximated secure activation function. At the end of secure inference, the server has learned nothing but the client learns the inference result. More details of cryptographic primitives can be found in Appendix A.1.

### 2.1 CRYPTOGRAPHIC INFERENCE.

Modern neural networks usually consist of linear convolution layers and non-linear activation layers. As Figure 1a shows, current state-of-the-art cryptographic inference, Delphi by Mishra et al. (2020),

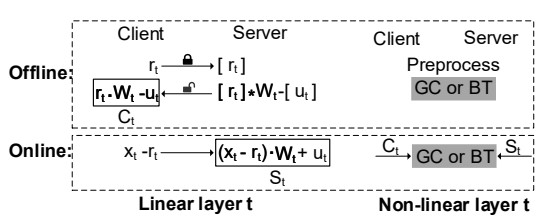

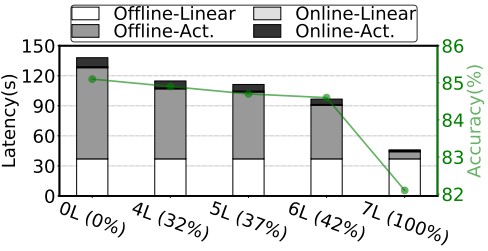

(a) The cryptographic inference in our baseline Delphi.    (b) Various approximate layers number (ratio).

Figure 1: (a) The secure inference scheme of CNN-7 on CIFAR-10. (b) Latency breakdown under activation approximation at a minimum of 84.5% accuracy. $X$L($Y$%) means $X$-layer activation or $Y$%-channel activation are approximated, e.g. 4L(32%) means 4-layer activation is approximated which equals 32% activation channels are approximated in the network.

has an offline phase and an online phase. The offline phase is independent of input data and is used to prepare data for the subsequent online phase. Each phase has linear and non-linear operations.

**1. Offline linear layer.** During the offline linear process, the client samples a random matrix $r_t$ that has the same shape with private input $x_t$, and then sends its encryption $[r_t]$ to server. The server processes homomorphic convolutions and returns $[r_t] * W_t - [u_t]$ to client, where $W_t$ is the $t$-$layer$ network weights and $[u_t]$ is a ciphertext of sampled matrix by server. The last step in the offline linear layer is that client obtains $C_t = r_t \cdot W_t - u_t$ which is one part of secret sharing of $x_t \cdot W_t$.

**2. Online linear layer.** The online linear phase aims to let the server obtain $S_t = (x_t - r_t) \cdot W_t + u_t$ which is the other secret sharing part of $x_t \cdot W_t$. It is almost as fast as the unencrypted computation, since the online input is a plaintext $x_t - r_t$.

**3. Offline Layer-Wise Activation layer.** Delphi supports a layer-wise activation function where each activation layer either is $ReLU$ based on GC or is the approximated polynomial based on BT. During the offline phase, GC needs to generate and share the garbled circuits. BT needs to generate and share the Beaver's triples.

**4. Online Layer-Wise Activation layer.** During the online phase of layer-wise activation, $ReLU$ is either performed by GC or the approximated degree-2 polynomial. The latency of approximated activation implemented by BT is 192× smaller than $ReLU$ based on GC.

**Latency Bottleneck and Motivation.** Figure 1b shows our baseline Delphi suffers from long latency and low accuracy under coarse-grained layer-wise activation approximation. Specifically, $0L$ in Figure 1b means none of the $ReLU$ layers and 0% $ReLU$ units are approximated by polynomials. The activation latency takes 72.5% of total latency. For only the online phase, activation latency occupies $\sim 99\%$ of the online latency. Therefore, activation layers are the performance bottleneck. $0L$ also shows the all-$ReLU$ model achieves 85.1% accuracy. With increased approximation layer numbers, Delphi is able to improve the approximation ratios, but the inference accuracy is decreased at the same time. With the 84.5% accuracy constraints, Delphi at most replaces 6-layer $ReLU$ layers, with an approximation ratio of $\sim 42\%$ $ReLU$ units. Figure 2 shows the reason why the approximation ratio is so low (42%); it is difficult to totally replace the $ReLU$ units by polynomials in the $bottleneck$ layer that has $> 58\%$ $ReLU$ units without a large decrease in accuracy. The $ReLU$ units in modern networks are mainly located in the first few layers, and the $ReLU$ numbers are usually decreased exponentially as shown in Figure 2. Especially for much deeper neural networks on a large dataset, replacing the first $ReLU$ layer significantly decreases the accuracy. To solve the above problems, we propose a more fine-grained channel-wise activation approximation method which is modelled as a hyper-parameter optimization problem.

## 2.2 POPULATION BASED TRAINING (PBT).

Inspired from evolutionary algorithms, Population Based Training (PBT) proposed by Jaderberg et al. (2017) is a more efficient method to jointly optimize model weights and user-specified hyper-parameters automatically during training. Many Reinforcement Learning (RL) based hyper-parameter optimization algorithms by Wang et al. (2019) and Lou et al. (2020) are not able to efficiently optimize hyper-parameters, since they simply stop training prematurely and consider partially trained accuracy as the final accuracy or reward. The details of PBT can be seen in ap-

pendix A.2 and PBT by Jaderberg et al. (2017). In this paper, we model the channel-wise activation approximation task as a hyper-parameter optimization problem.

| Features | TAPAS | Gazelle | NASS | CONAD | Delphi | CryptoNAS | InstaHide | **SAFENet** |
|---|---|---|---|---|---|---|---|---|
| Strong Encryption | ✓ | ✓ | ✓ | ✓ | ✓ | ✓ | ✗ | ✓ |
| Batched HE | ✗ | ✓ | ✓ | ✓ | ✓ | ✓ | - | ✓ |
| Optimized Activation | ✓ | ✗ | ✗ | ✓ | ✓ | ✓ | - | ✓ |
| Channel-Wise | ✗ | ✗ | ✗ | ✗ | ✗ | ✗ | - | ✓ |
| Mixed-Precision | ✗ | ✗ | ✗ | ✗ | ✗ | ✗ | - | ✓ |
| Multiple-Degrees | ✗ | ✗ | ✗ | ✗ | ✗ | ✗ | - | ✓ |

Table 1: Cryptographic inference works.

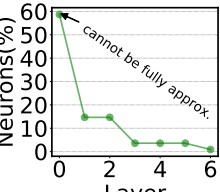

Figure 2: Neurons ratio.

### 2.3 COMPARISON WITH PRIOR WORKS.

Table 1 shows a comparison between prior works and SAFENet. TAPAS by Sanyal et al. (2018), XONN by Riazi et al. (2019) and soteria by Aggarwal et al. (2020) focus on binary neural networks which uses $sign()$ function instead of the $ReLU$ activation, thereby suffering from inference accuracy loss. And TAPAS by Sanyal et al. (2018) and SHE by Lou & Jiang (2019) suffer from long-latency linear operations since they adapt a HE scheme called TFHE by Chillotti et al. (2018) that does not support ciphertext batching operations yet. For example, TAPAS and SHE take $\sim 2$ hour and $\sim 10$ seconds respectively to perform one single MNIST inference with 99% accuracy. In contrast, our work SAFENet and Gazelle using the hybrid of batched HE and MPC are able to achieve $< 1$-second latency with $> 99\%$ accuracy. Gazelle by Juvekar et al. (2018) and MiniONN prove the feasibility of the hybrid use of GC and HE, but they both suffer from enormous latency. NASS by Bian et al. (2020), CONAD by Shafran et al. (2019), and CryptoNAS by Ghodsi et al. (2020) try to improve Gazelle and MiniONN's performance by the co-design of neural network architectures and cryptographic protocol, but they all require a heavy, online, cryptographic phase. Delphi by Mishra et al. (2020) reduces online latency by moving some online operations into the offline phase and replacing layer-wise activation by degree-2 polynomials. Only our SAFENet supports more fine-grained, channel-wise, activation approximation with multiple-degree polynomial exploration, shown in Table 1. SAFENet also enables the mixed-precision approximation ratios for different layers. Other than MPC and HE based neural networks, InstaHide by Huang et al. (2020b) and TextHide by Huang et al. (2020a) use a class of subset-sum type encryption by Bhattacharyya et al. (2011) to protect the user's sensitive data in machine learning service with only $< 5\%$ computational overhead and little accuracy loss. However, an attack by Carlini et al. (2020) shows that there is a potential security risk on the InstaHide. Compared to the light-weight methods InstaHide and TextHide, MPC and HE provide much stronger security guarantees.

## 3 SAFENET

**Overview.** In order to replace expensive non-linear activations by cheap linear polynomial approximations as much as possible, we mainly use three methods. Firstly, we use fine-grained activation approximation in a channel-wise manner, instead of coarse-grained approximation in a layer-wise manner. Secondly, since different layers in neural networks have different contributions on the inference accuracy, we propose an automatic mixed-precision method to assign a proper approximation ratio for each layer. According to the assigned approximation ratio, SAFENet replaces the corresponding channels in the ascending order of their channel important factors. Mixed-precision further improves the total approximation ratio, given an accuracy threshold. Moreover, SAFENet's mixed-precision planner explores different polynomial approximations of various degrees.

**SAFENet Methods Definition.** Given an $T$-Layer CNN model, we define that $\{W_t\}_{t=0}^{T-1}$ and $\{A_t\}_{t=0}^{T-1}$ are the model weights and activations, and let $W_t \in \mathbb{R}^{c_{out} \times c_{in} \times w_w \times h_w}$ and $A_t \in \mathbb{R}^{c_{in} \times w_a \times h_a}$ be the $t$-th layer model weights and activations, where $c_{out}$ is output channel number, $c_{in}$ is input channel number, $w_w$ and $h_w$ are kernels' width and height, $w_a$ and $h_a$ are activations' width and height. The $t$-th layer output feature maps $O_t \in \mathbb{R}^{c_{out} \times w_o \times h_o}$ can be derived by Equation 1, where $O_{t,i,:,:}$ is the output feature map of $t$-th layer $i$-th channel, $A_{t,j,:,:}$ is the $t$-th layer and $j$-th channel activation, $W_{t,i,j,:,:}$ is the model parameters of $t$-th layer, $i$-th output channel and $j$-th input channel, $w_o$ and $h_o$ are output feature's width and weight, and $*$ is the convolution operation.

$$O_{t,i,:,:} = \sum_{j=1}^{c_{in}} A_{t,j,:,:} * W_{t,i,j,:,:}. \tag{1}$$

Then the channel-wise activation approximation can be defined as Equation 2, where $(t+1)$-th layer activation $A_{t+1}$ of $i$-th channel is derived by expensive $ReLU$ when the corresponding mask matrix, $m_{t+1,i} = 0$, otherwise the approximated activation.

$$A_{t+1,i,:,:} = \begin{cases} ReLU(O_{t,i,:,:}), & \text{if } m_{t+1,i} = 0. \\ Approx(O_{t,i,:,:}), & \text{if } m_{t+1,i} = 1. \end{cases} \tag{2}$$

$$m_{t+1,i} = \begin{cases} 0, & \text{if } imp\_order(A_{t+1,i}) \geq n_{t+1} \times (1 - \alpha_{t+1}). \\ 1, & \text{Otherwise.} \end{cases} \tag{3}$$

Equation 3 defines the mask matrix $m_{t+1,i}$. Here, $imp\_order(A_{t+1,i}) \in [1, c_{out}]$ is the importance ascending order of $i$-th channel $A_{t+1,i}$ in the layer $(t+1)$, and $imp\_order(A_{t+1,i})$ can be derived by $sort(\sum_{j=1}^{C_{in}} |W_{t,i,j,:,:}|)$, where $\sum_{j=1}^{C_{in}} |W_{t,i,j,:,:}|$ is the $L$-1 norm of $i$-th channel weight, and this $L$-1 norm is proven effective as the channel importance of $A_{t+1,i}$ by prior works by Li et al. (2017) and by Liu et al. (2019). $n_{t+1}$ in Equation 3 is the neuron number in $(t+1)$-th layer and $\alpha_{t+1}$ is the activation approximation ratio in the $(t+1)$-th layer. Equation 3 shows that channel-wise activation approximation in the $(t+1)$-th layer only approximate $\alpha_{t+1}$-ratio less important activations with polynomials by setting their corresponding mask matrix entry $m_{t+1,i} = 1$. $Approx()$ activation function is enabled by multiple options including degree-3 polynomial $a_1 x^3 + a_2 x^2 + a_3 x + a_4$, degree-2 polynomial $b_1 x^2 + b_2 x + b_3$, and degree-0 pruning, where $a_1$ to $a_4$ and $b_1$ to $b_3$ values are derived from the model training. The reason why we choose these three options is that our experiments show larger degree $(> 3)$ polynomial approximation is hardly convergent in training because of the uncontrollable gradient exploding, and smaller degree $(< 2)$ polynomial approximation shows worse performance. However, the proper insertions of degree-0 pruning may help on gradient exploding in some layers. We use hyper-parameter $\beta_t \in \{0, 2, 3\}$ as the polynomial degree in the $Approx()$ function. **Channel-Wise Activation Approximation** means $T$ activation approximation ratios $\{\alpha_t\}_{t=0}^{T-1}$ are designed in a hand-crafted manner with $\beta_t = 2$. **Mixed-Precision Activation Approximation** automatically search the optimal combination of $\{\alpha_t\}_{t=0}^{T-1}$ using our improved PBT algorithm called BTPBT. BTPBT is described in Algorithm 1. **Multiple-degree approximation** means that approximation options $\{\beta_t\}_{t=0}^{T-1}$ are searched with $\alpha_t$ by BTPBT algorithm.

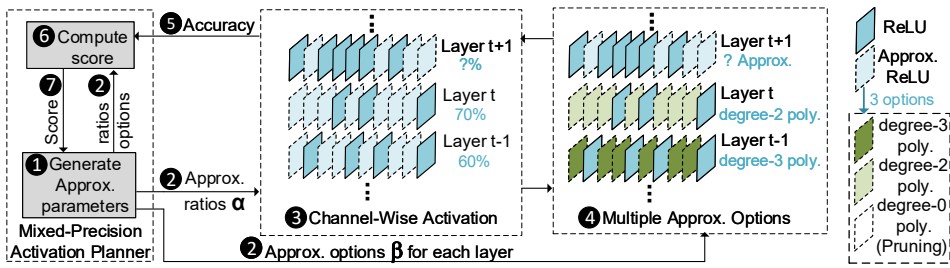

Figure 3: The overview of SAFENet working scheme. SAFENet is used to search optimal $\alpha_t$ and $\beta_t$ to get best score in the training. $\alpha_t$ decides the approximation channels in the $t$-th layer, $\beta_t \in \{0, 2, 3\}$ is the polynomial degree in the $Approx()$ function.

**Working Flow.** Figure 3 depicts the working flow of SAFENet. ❶ At first, the mixed-precision activation planner randomly initiates approximation parameters, $\{\alpha_t\}_{t=1}^{T-1}$ and $\{\beta_t\}_{t=1}^{T-1}$, to control channel-wise activation approximation and approximation options for each layer. ❷ Approximation parameters including $\{\alpha_t\}_{t=1}^{T-1}$ and $\{\beta_t\}_{t=1}^{T-1}$ are received by the neural network model to decide which activation channels are approximated first, and then to decide which approximation method is picked up. ❸ In each $t$-th layer, $n_t \cdot \alpha_t$ unimportant channels are approximated by polynomials and $n_t \cdot (1 - \alpha_t)$ more important channels are kept as original activation function, e.g. $ReLU$, according to Equation 2 and Equation 3. ❹ The approximation methods for $n_t \cdot \alpha_t$ unimportant $ReLU$ units are decided. The approximation option consists of multiple-degree polynomials, including degree-$\beta_t$ polynomials, where $\beta_t \in \{0, 2, 3\}$. ❺ The neural network model with channel-wise activation approximation is performed for several times and outputs its inference accuracy. ❻ The mixed-precision activation planner uses the BTPBT method to compute the optimizing score defined in Equation 4, aware of accuracy and reduced latency at the same time. ❼ The generated score is used

to guide the approximation parameters in the next iteration. These steps repeat until cryptographic inference accuracy and latency threshold are satisfied.

**Score.** The goal of our SAFENet is to reduce cryptography latency by replacing expensive original activation with cheap approximated activation as much as possible, given the user-defined accuracy threshold. Equation 4 is used to assign a *score* for the channel-wise activation approximation parameters $\alpha_t$ and $\beta_t$. $\sum_{t=1}^{T} \alpha_t \times n_t$ is the number of $ReLU$ units that are replaced. $R(\beta_t)$ means the reduced time ratio using a degree-$\beta_t$ polynomial to replace $ReLU$ units. For example, $R(\beta_t = 3) = 1 - \frac{0.2}{20.1} = 99.9\%$, since amortized degree-3 polynomial and $ReLU$ cost 0.2 $us$ and 20.1 $us$, respectively. Therefore, $\sum_{t=1}^{T} \alpha_t \times n_t \times R(\beta_t)$ represents the reduced ratio of total latency. Our *score* keeps the balance of accuracy and efficiency, enabling users to replace more activation neurons with multiple approximated polynomial degrees, while meeting the accuracy constraints.

$$score = accuracy \times (1 + \sum_{t=1}^{T} \alpha_t \times n_t \times R(\beta_t)) \qquad (4)$$

---

**Algorithm 1** Binary Tree Population Based Training BTPBT($M_T, D, A_t, W$)

---

**Input:** A $T$-layer neural network $M_T$ with weight $W$, training data $D$, accuracy threshold $A_t$.
**Output:** Activation approximation rate $\alpha[0 : T-1]$, polynomial degree $\beta[0 : T-1]$, score $S$, accuracy $A$ and weight $W$.
1: Construct a binary tree, the root node $root\_nd$ contains $layers[0 : T-1]$ and all layers in the $root\_nd$ share the same $\alpha$ and $\beta$.
2: Search one single $\alpha$ and $\beta$ for $root\_nd$ using PBT($\alpha, \beta, D, W$). Add $root\_nd$ into an empty queue $root\_queue$.
3: **while** $root\_queue$ is not **Null do**
4:      $parent\_nd = root\_queue.pop()$;
5:      $left\_child, right\_child = $ BinaryDivide($parent\_nd$);
6:      $(\alpha[0 : T-1], \beta[0 : T-1], S, A, W) = $ PBT($left\_child.\{\alpha, \beta\}, right\_child.\{\alpha, \beta\}, D, W$);
7:      $root\_queue.push(left\_child)$ if $left\_child$ has $> 1$ layer; Same operation on $right\_child$;
8: Pick up the intermediate result $(\alpha[0 : T-1], \beta[0 : T-1], S, A, W)$ as the best candidate $BC$ that has the the largest score $S$ and accuracy $A > A_t$ .
9: Perform $(\alpha[0 : T-1], \beta[0 : T-1], S, A, W) = $PBT($BC.\{\alpha[0 : T-1], \beta[0 : T-1]\}, D, W$);
10: **Return** $\alpha[0 : T-1], \beta[0 : T-1], S, A, W$

---

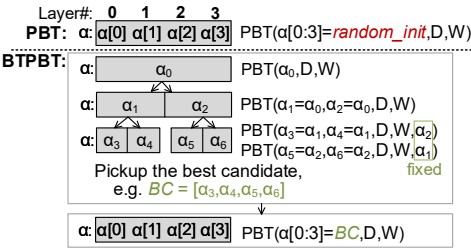
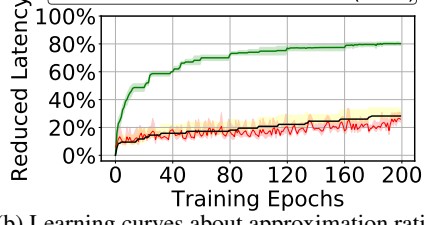

(a) BTPBT example for a 4-layer network.     (b) Learning curves about approximation ratio.

Figure 4: (a) Comparison of PBT and BTPBT. (b) The learning curve of CNN-7 on CIFAR-10.

**Mixed-Precision Activation Planner.** Finding appropriate parameters $\alpha_t$ and $\beta_t$ to achieve a higher score in Equation 4 is a typical hyper-parameter optimization problem. Hand-crafted optimization for parameters is only practical to handle the search space of a single pair of $\alpha$ and $\beta$ for all layers, which confines the improvement effect on activation approximation. Therefore, mixed-precision activation approximation is proposed to enable each layer to have unique $\alpha_i$ and $\beta_i$. However, applying hyper-parameter optimization methods to automatically search mixed-precision activation approximation is non-trivial. This is because of the enormous search space of $\alpha_t$ and $\beta_t$ for each layer. Assume $\alpha_t \in [0, 1]$ and it is discretized into the times of 0.05, therefore 21 options; $\beta_t \in \{0, 2, 3\}$, has 3 options. If a neural network model has $T$ layers, the search space is $(3 \times 21)^T$. For this large search space, even PBT by Jaderberg et al. (2017) and Deep Reinforcement Learning (DRL) based methods by Wang et al. (2020) suffer from slow learning speed and low approximation ratio. Figure 4b shows PBT and DRL methods cannot effectively reduce latency by mixed-precision activation approximation.

We propose a Binary-Tree PBT algorithm (BTPBT) in Algorithm 1 to enable the hyper-parameters search of $\alpha_t$ and $\beta_t$. BTPBT in Algorithm 1 takes the $T$-layer neural network $M_T$, training data $D$

and accuracy threshold $A_t$ as inputs, and returns all the searched $\alpha_t$ and $\beta_t$ with updated weights $W$, accuracy $A$ and score $S$. Instead of using PBT to search all $\alpha[0 : T - 1]$ and $\beta[0 : T - 1]$ directly, BTPBT starts from an easier task that limits all $\alpha[0 : T - 1]$ sharing the same value, e.g. $\alpha$, and all $\beta[0 : T - 1]$ sharing the same value, e.g. $\beta$. This searching process about $root\_nd$ is shown in the line 1 of Algorithm 1. Then the $root\_nd$ with all $T$ layers is equally divide into two child nodes using $BinaryDivide()$ function, so that the left child has the first-half layers of $root\_nd$ and the right child has the remaining layers. We then perform PBT on both left and right child nodes, and limit all layers in each child share the same $\alpha$ and $\beta$. Here, the total approximated activation channels of two child nodes should be larger than the approximated channels of the parent node. The child nodes will become new root nodes and PBT will be performed repeatedly on sibling nodes until the $root\_queue$ is null in Algorithm 1. The PBT operations in the binary tree can also be used for the warm-starting initialization for the joint search of all $\alpha[0 : T - 1]$ and $\beta[0 : T - 1]$ shown in line 8~9 of Algorithm 1.

Figure 4a shows a comparison of PBT and BTPBT for a $T$-layer network ($T = 4$). PBT directly performs a joint search on $\alpha[0 : T - 1]$ and $\beta[0 : T - 1]$ (We skip the $\beta$ in Figure 4a due to space limitation). The enormous search space and random initialization of $\alpha[0 : T - 1]$ and $\beta[0 : T - 1]$ make PBT difficult to learn useful information in training. Therefore directly using PBT suffers from slow learning speed and inadequate searching results. To solve this issue, we use BTPBT to firstly search one single $\alpha_0$ for all layers and then search more $\alpha$ iteratively in a binary tree order. In this way, the intermediate results about $\alpha$, $\beta$ and weights $W$ can be used for the final results or the candidates of the further search. For instance, the intermediate result with the largest score $S$ can be considered as the best candidate for the warm-starting initialization for the joint search of $\alpha[0 : T - 1]$. More details about BTPBT can be seen at Appendix A.3. Figure 4b depicts the learning curves of PTB, DRL by Wang et al. (2020), and our SAFENet. All these methods are used to search hyper-parameters $\alpha_t$ and $\beta_t$ for the same score shown in Equation 4. After 200 training epochs, BTPBT is able to replace $\sim 80\%$ neurons, but PBT and DRL only replace $< 30\%$ neurons. In the beginning of training, our baselines and BTPBT are given the same pre-trained model. The DRL setting is specified in Wang et al. (2020). For PBT and our work, the evolution cycle between two exploitation is 20 iterations, and 50 workers are used to simultaneously search parameters. For BTPBT, each two sibling nodes and the root node require a PBT separately, therefore $2^{levels}$ PBT is required. We set 20 epochs for each node's PBT. We use 40 epochs to jointly search the activation approximation. The total 200 epochs with 50 workers take $\sim 7$ GPU hours.

**Threat model of BTPBT planner.** Our BTPBT takes the all-$ReLU$ neural network model $M_T$ and training data $D$ as inputs in the server side, and outputs a well-trained polynomial-approximated neural network $P_T$. The $P_T$ is used to provide a privacy-preserving machine learning inference service that takes the client's encrypted sensitive data $D_{inf}$ as input, and outputs the encrypted prediction result to the client. So our BTPBT does NOT impact the privacy of user's data $D_{inf}$. BTPBT shares the same threat model with our baseline Delphi by Mishra et al. (2020).

## 4 EXPERIMENTAL METHODOLOGY

**Benchmarks and Dataset.** Our secure inference experiments rely on CIFAR-10 and CIFAR-100 datasets. Both CIFAR-10 and CIFAR-100 contain 50000 training images and 10000 testing images, where each image size is $32 \times 32 \times 3$. Images in CIFAR-10 are classified into 10 classes, but CIFAR-100 has 100 classes. Our baseline Delphi shows that CIFAR-100 is the most difficult dataset in the existing secure inference. We adopt CNN-7 specified in MiniONN by Liu et al. (2017) to evaluate CIFAR-10. This CNN-7 architecture is also used in Delphi by Mishra et al. (2020). Moreover, a deeper network VGG-16 by Geifman (2017) is evaluated to further improve inference accuracy on CIFAR-10. For CIFAR-100, ResNet-32 by He et al. (2016) is evaluated.

**Systems Setup.** We ran the secure inference models on two instances. They are equipped with an Intel Xeon E7-4850 CPU and 16 GB DRAM. The communication links between these two instances are in the LAN setting, and each instance uses 4 threads, same as the previous works. The hyper-parameters optimization for activation replacement requires an NVIDIA Tesla V100 GPU. The SEAL library by SEAL and Multi-Protocol fancy-garbling library by Carmer et al. (2019) are used to implement HE and garbled circuits functions. To avoid overflow, our experiments adopt 15-bit fixed point representation and the Least Significant Bits of intermediate results are also truncated

to 15 bits. The hyper-parameters selection is implemented in Python, and the BTPBT is constructed based on regular PBT by Jaderberg et al. (2017) in Tune platform by Liaw et al. (2018).

## 5 RESULTS

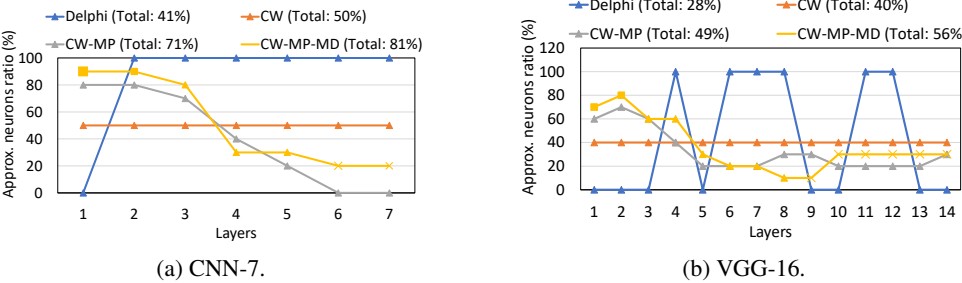

(a) CNN-7.           (b) VGG-16.

Figure 5: Approximated ratio ablation study on CIFAR-10 at a minimum of 84.5% accuracy.

**Ablation Effects.** Figure 5 describes the ablation effects of proposed techniques on CIFAR-10 with CNN-7 and VGG-16. CW, MP, and MD mean Channel-Wise, Mixed-Precisicion and Multiple Degrees, respectively. Square, triangle and cross marks in Figure 5 represent degree-3, degree-2 polynomial approximation and pruning, respectively. For the CNN-7 in Figure 5a, our baseline Delphi approximates $\sim 41\%$ activation by replacing the $ReLU$ in the last 6 layers with degree-2 polynomials. Channel-wise (CW) activation approximation using $\alpha = 0.5$ and $\beta = 2$ on all layers is able to achieve the same inference accuracy, while replacing more expensive activation neurons. Channel-wise under mixed precision (CW-MP) further improves the approximation ratio of the regular channel-wise method. Other than channel-wise approximation and mixed precision, Figure 5a shows that the approximation ratio also benefits from exploring multiple degrees in various layers (CW-MP-MD). Figure 5b depicts the ablation study of a deeper network VGG-16. Our SAFENet with CW-MP-MD improves the approximation $ReLU$ numbers of Delphi by 28% under the same accuracy.

| CNN-7 | Offline | | | Online | | | Total | Accuracy |
|---|---|---|---|---|---|---|---|---|
| | HE-Linear | MPC-Act | Total | HE-Linear | MPC-Act | Total | | |
| Gazelle | 0s | 91s | 91 s | 37s | 9s | 46s | 137s | 85.1% |
| Delphi | 37s | 53.4s | 90.4s | 0s | 5.3s | 5.3s | 95.4s | 84.6% |
| Delphi-Fast | 37s | 15.1s | 52.1s | 0s | 1.69s | 1.69s | 53.7s | 83.1% |
| **SAFENet** | **33.3s** | **18.2s** | **51.6s** | **0s** | **1.8s** | **1.8s** | **53.4s** | **85.1%** |

Table 2: The CNN-7 result on CIFAR-10. Delphi-Fast approximates more activation than Delphi, but losses more accuracy.

| VGG-16 | Offline | | | Online | | | Total | Accuracy |
|---|---|---|---|---|---|---|---|---|
| | HE-Linear | MPC-Act | Total | HE-Linear | MPC-Act | Total | | |
| Gazelle | 0s | 142s | 142s | 45s | 16s | 61s | 203s | 89.6% |
| Delphi | 45s | 114s | 159s | 0s | 13s | 13s | 172s | 88.1% |
| Delphi-Fast | 45s | 109s | 154s | 0s | 12.4s | 12.4s | 166s | 84.9% |
| **SAFENet** | **40.2s** | **56.8s** | **97s** | **0s** | **6.5s** | **6.5s** | **104s** | **88.9%** |

Table 3: The VGG-16 result on CIFAR-10. Delphi-Fast approximates more activation than Delphi, but losses more accuracy.

**CIFAR-10.** Table 2 shows the comparisons of CNN-7 implemented by prior techniques and SAFENet on CIFAR-10. Compared to Gazelle, Delphi not only moves the HE-based linear operations to the offline, but also reduces $\sim 41\%$ online and offline activation latency without significant accuracy loss. Our work SAFENet has similar latency with Delphi in the linear layer, but SAFENet remarkably reduces latency at non-linear activation layers. More specifically, SAFENet eliminates $\sim 66\%$ online and offline latency in activation layers, and reduces $\sim 61\%$ total latency with higher accuracy than Delphi. Delphi-Fast further reduces the latency of Delphi by trying to approximate more activation than Delphi, but it suffers from a significantly accuracy decrease. For

instance, Delphi-Fast decreases 2% accuracy when it has similar latency with SAFENet. We also evaluated a deeper neural network VGG-16 on CIFAR-10 shown in Table 3. When Gazelle and Delphi have similar accuracy, Delphi only reduces $< 20\%$ latency of Gazelle on non-linear activation layer. This shows that Delphi benefits less from deeper neural networks. Compared to Gazelle, our work SAFENet is able to reduce $\sim 59.4\%$ online latency, 48.7% total latency. SAFENet has 50% less online latency and 39.5% less total latency with higher accuracy than Delphi, since SAFENet uses more find-grained channel-wise activation approximation. Compared to Delphi-Fast, SAFENet achieves 4% accuracy improvement with a less inference latency.

| ResNet-32 | Offline | | | Online | | | Total | Accuracy |
|---|---|---|---|---|---|---|---|---|
| | HE-Linear | MPC-Act | Total | HE-Linear | MPC-Act | Total | | |
| Gazelle | 0s | 158s | 158s | 64s | 18s | 82s | 240s | 67.9% |
| Delphi | 64s | 126.4s | 190.4s | 0s | 14.4s | 14.4s | 204.8s | 67.3% |
| Delphi-Fast | 64s | 121.1s | 175.1s | 0s | 13.8s | 13.8s | 198.9s | 65.7% |
| **SAFENet** | **57.6s** | **63.2s** | **120.8s** | **0s** | **7.2s** | **7.2s** | **128s** | **67.5%** |

Table 4: CIFAR-100 results using ResNet-32. Delphi-Fast approximates more activation than Delphi, but losses more accuracy.

**CIFAR-100.** Table 4 shows the results of ResNet-32 evaluated on CIFAR-100. Gazelle achieves 67.9% inference accuracy, and each image inference consists of a 158-second offline phase and a 82-second online phase. Delphi is faster than Gazelle's online latency by 67.6 seconds with only a 0.4% accuracy decrease, but increases the offline latency by $\sim 30$ seconds. The offline phase is enlarged because Delphi moves most online HE operations to offline phase and only reduces few GC operations offline. Compared to Gazelle, our work SAFENet is able to reduce both online and offline latency since we significantly reduce the number of GC-based activations; SAFENet reduces online latency by 91.2% and offline latency by 23.5%. Compared to Delphi, SAFENet has 50% less online latency and 36.6% less offline latency. SAFENet reduces total latency by 46.7% and 37.5% over Gazelle and Delphi, respectively. In addition, Delphi suffers from a significant accuracy decrease when trying to approximate more activations. For example, Delphi-Fast approximates one more layer, losing 1.6% accuracy. Our work SAFENet improves 1.8% inference accuracy and reduces 35.8% latency over Delphi-Fast.

## 6 CONCLUSION

In this paper, we propose SAFENet to enable a Fast, Accurate and Secure neural Network inference service. SAFENet consists of three techniques including channel-wise activation, multiple-degree polynomial approximation, and mixed-precision approximation ratios. The channel-wise activation approximation keeps the most useful activation channels and replaces the remaining less useful channels with the various-degree polynomials. The mixed-precision activation approximation is implemented by assigning various layers with different approximation ratios further increasing the approximation ratio and reducing inference latency. Our experimental results show SAFENet obtains the state-of-the-art inference latency and performance, decreasing latency by $38\% \sim 61\%$ or improving accuracy by $1.8\% \sim 4\%$ over prior techniques.

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

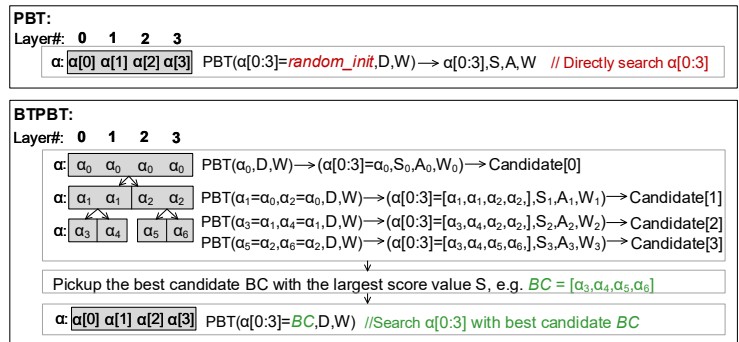

Figure 6: An comparison of PBT and our BTPBT on the $\alpha$ search of a 4-layer network.

# A APPENDIX

## A.1 CRYPTOGRAPHIC PRIMITIVES.

**Homomorphic Encryption (HE).** Homomorphic Encryption (HE) is a cryptosystem that supports computation on ciphertexts $x$ without decryption or private key. Given public key $pk$, private key $sk$, encryption function $\epsilon()$, and decryption function $\sigma()$, we can define an homomorphic operation $\times$ if there is an operation $\otimes$ such that $\sigma(\epsilon(x_1, pk) \otimes \epsilon(x_2, pk), sk) = \sigma(\epsilon(x_1 \times x_2, pk), sk)$, where $x_1$ and $x_2$ are sensitive plaintext from data holders, and only data holders have the private key $sk$.

**Secret Sharing (SS) and Garbled Circuit (GC).** Secret Sharing (SS) is usually based on additive sharing private values between the parties. For example, one variable $x$ can be shared into two parts: $\langle x \rangle_1^A$ in $P_1$ and $\langle x \rangle_2^A$ in $P_2$, and variable $x$ can be reconstructed by $x = \langle x \rangle_1^A + \langle x \rangle_2^A$. In previous secure inference schemes by Juvekar et al. (2018), SS is used to protect the privacy of intermediate results in the hidden layers. Garbled Circuit (GC) is a cryptographic protocol that enables two parties (Garbler and Evaluator) to jointly compute a function over their private data without learning the other party's data. To use GC, the computed function should be represented into a Boolean circuit. The Garbler firstly garbles the Boolean circuit and generates the garbled table. The Evaluator receives the garbled table by the Oblivious Transfer by Juvekar et al. (2018) and evaluates the table to the result. Previous secure inference schemes by Juvekar et al. (2018) show that GC is used to process non-linear activation like $ReLU$ function.

**Beaver's multiplicative Triples (BT).** Beaver's multiplicative Triples proposed by Beaver (1995) can be considered as a secure two-party computation protocol. Assume two parties $P_1$ and $P_2$ have variables $x$ and $y$, respectively. BT protocol enables two parties to obtain the secret sharing of product $xy$ without revealing $x$ and $y$. More details can be seen in Delphi by Mishra et al. (2020). BT-based polynomial approximation for $ReLU$ is 3-order magnitude cheaper than GC-based $ReLU$ units on average, so it is used to design approximated secure activation function.

## A.2 POPULATION BASED TRAINING(PBT).

PBT is computationally efficient because it adapts parallel scheme and weight sharing during the evolutionary process. Specifically, given a pre-trained model, many workers are created and each worker shares the same pre-trained weights and has unique hyper-parameters. Multiple workers are then independently trained for several iterations and evaluated by the user-specified score function. In the exploitation, workers with better scores will keep their parameters, and copy their weights and hyper-parameters to the other workers. Workers with worse scores will perform exploration to randomly scale their hyper-parameters. The training, scoring, exploitation and exploration repeat until the score function is convergent.

## A.3 AN COMPARISON OF PBT AND BINARY-TREE PBT(BTPBT).

Figure 6 shows a comparison of PBT and BTPBT for a $T$-layer network ($T = 4$). PBT directly performs a joint search on $\alpha[0:3]$ and $\beta[0:3]$. The enormous search space and random initializa-

tion of $\alpha[0:3]$ and $\beta[0:3]$ make PBT difficult to learn useful information in training. Therefore directly using PBT suffers from slow learning speed and inadequate searching results. To solve this issue, we use BTPBT to firstly search one single $\alpha_0$ for all layers and then search more $\alpha$ iteratively in a binary tree order. The root node $root\_nd$ of the binary tree is constructed by assigning a single $\alpha_0$ to all the 4 layers in the neural network. One PBT is performed to search the $\alpha_0$, given the dataset $D$ and initialized weights $W$. The returning result including $\alpha_0$, score $S$, accuracy $A$ and trained weights $S$ is called candidate[0]. After that, all layers in the $root\_nd$ are equally divide into two parts: $left\_child$ and $right\_child$. The $left\_child$ contains the first two layers which shares the same $\alpha_1$, and the $right\_child$ includes the last two layers which shares the same $\alpha_2$. Then one single PBT is used to search $\alpha_1$ and $\alpha_2$ with the score $S$, accuracy $A$ and trained weights $W$. The returning, intermediate results are called candidate[1]. The search of $left\_child$ node is much more important since the former layers in the neural network contains more neuron activations than the latter layers. This BTPBT searching process is repeated until each node has only one single layer. The layer number of binary tree is equal to $log_2 T + 1$, where $T$ is the layer number of neural network. In this way, the intermediate results about $\alpha$, $\beta$ and weights $W$ can be used for the final results or the candidates of the further search. For instance, the intermediate result with the largest score $S$ can be considered as the best candidate $BC$ for the warm-starting initialization for the joint search of $\alpha[0:3]$. Our BTPBT is able to reduce the training time and improve the secure inference performance over the prior PBT.

