# OpenReview forum: "SAFENet: A Secure, Accurate and Fast Neural Network Inference"
_ICLR.cc/2021/Conference — ICLR 2021 Poster_

### Official Review · AnonReviewer2 · 2020-10-25
**An ok submission, but writing quality requires more improvement**

**Rating:** 5
**Confidence:** 3

**Review:**




This paper proposed a new method called SAFENet.  It is able to support a secure, accurate and fast neural network inference service. The best result presented in Table 2, is 3x faster than previous. The best result presented in Table 2, is 2x faster than previous.

Algorithm 1 is not very well written, it is hard to follow what is that algorithm trying to say.

In page 2, the 2nd line in Section 2.1, the sentence ``current state-of-the-art cryptographic inference, Delphi'' It should have a citation here. The section 2 is related work, but Section 2.1 doesn't cite any previous papers, this is a bit strange.

In the introduction, this paper discusses about cryptography but many important references are missing. For example [Yao] and [Gentry]

Andrew Yao. Protocols for secure computations. FOCS 1982.

Criag Gentry. Fully homomorphic encryption using ideal lattices. STOC 2009.


In related work, the following recent work should be discussed, since they had proposed a practical way to encrypt the image and text data.

Yangsibo Huang, Zhao Song, Kai Li, Sanjeev Arora. InstaHide: instance-hiding schemes for private distributed learning. ICML 2020
Yangsibo Huang, Zhao Song, Danqi Chen, Kai Li, Sanjeev Arora. TextHide: Tackling Data Privacy in Language Understanding Tasks. EMNLP 2020

---

> ### Author Response · Authors · 2020-11-15
> **Reply to Reviewer 2**
>
> #### We would like to thank reviewer 2 for the thoughtful comments and efforts towards improving our manuscript.
>
>
> ##### Question 1. Algorithm 1 is not well-written and it is hard to follow.
>
> Answer 1: We sincerely appreciate this instructive review. We firstly re-write the Algorithm 1 and its description. We also take the updated figure 4a  and more detailed figure 6 as an example to better illustrate our Algorithm 1.  These updates can be seen in the revised submission now.  Here we introduce the context, motivation, function, and effect of our Algorithm 1.
>
> The context of our Algorithm 1(BTBPT) is that we model the channel-wise activation approximation as a hyper-parameters ($\alpha_t$ and $\beta_t$) search problem. However, the search space is too enormous so that Deep-Reinforcement Learning (DRL) and Polulation Based Training (PBT) suffer from slow learning speed and inadequate search performance.  To tackle this problem, we propose Algorithm 1 to better search the hyper-parameters. The Algorithm 1 takes the $T$-layer neural network $M_T$, training data $D$ and accuracy threshold $A_t$ as inputs, and outputs all the the $\alpha_t$ and $\beta_t$ with updated weights $W$, accuracy $A$ and score $S$.  Figure 4(a) shows a comparison of PBT and BTPBT for a $T$-layer network ($T=4$). Figure 4(b) depicts the learning curves of PTB, DRL, and our SAFENet, which are used to search hyper-parameters $\alpha_t$ and $\beta_t$ for the same score. After 200 training epochs, BTPBT is able to replace $ 80$ % neurons, but PBT and DRL only replace $<30$% neurons.
>
>
> ##### Question 2. Citation in the section 2. More citation about HE and MPC.
> Answer 2: We really appreciate reviewer 2 for the thoughtful comments towards improving our manuscript. We have fixed them in the revised manuscript.
>
> ##### Question 3. More related work discussion on InstaHide[1] and TextHide[2].
>
> Answer 3: We thank the reviewer for pointing these related works. We have added them into the related works in the revised manuscripts.  InstaHide[1] and TextHide[2] use a class of subset-sum type encryption[3] to protect the user's sensitive data in machine learning service with only $<5$% computational overhead and little accuracy loss. However, an attack[4] shows that there is a potential security risk on the InstaHide. The light-weight InstaHide and TextHide have advantages on performance (especially the latency), but Yao’s Gabled Circuits[5] and HE[6] are of advantages in the security guarantees so far.
>
> [1] Huang, Yangsibo, Zhao, Song, Kai, Li, and Sanjeev, Arora. "TextHide: Tackling Data Privacy in Language Understanding Tasks." . In International Conference on Machine Learning (ICML).2020.
>
> [2] Yangsibo Huang, Zhao Song, Danqi Chen, Kai Li, Sanjeev Arora. TextHide: Tackling Data Privacy in Language Understanding Tasks. EMNLP 2020
>
> [3]Bhattacharyya, Arnab, Piotr, Indyk, David P, Woodruff, and Ning, Xie. "The Complexity of Linear Dependence Problems in Vector Spaces.." . In ICS (pp. 496–508).2011.
>
> [4] Carlini, N., Samuel Deng, S. Garg, S. Jha, Saeed Mahloujifar, Mohammad Mahmoody, Shuang Song, Abhradeep Thakurta and Florian Tramèr. “An Attack on InstaHide: Is Private Learning Possible with Instance Encoding?” (2020).
>
> [5] Andrew Yao. Protocols for secure computations. FOCS 1982.
>
> [6] Criag Gentry. Fully homomorphic encryption using ideal lattices. STOC 2009.

---

### Official Review · AnonReviewer3 · 2020-10-28
**Further exploration of a known space in two-party secure inference**

**Rating:** 7
**Confidence:** 3

**Review:**

Summary:
The paper present a system for two-party deep learning inference. The main contribution is activation layers that are more expensive in two-party computation are replaced by approximations dynamically based on the training data. To this end, the authors use a divide-and-conquer approach to gauge the impact of replacing activation functions of some layers by a version more amenable to secure computation. Furthermore, the algorithm also considers various degrees for approximation (0, 2, and 3). Experiments show that this reduces the latency by up to two thirds while maintaining a similar accuracy.

Pros:
- The dynamic approach is more sophisticated.
- The improvement in efficiency is clear.

Cons:
- Deriving public meta-parameters from data might impact privacy, but no consideration is given to this aspect.
- The further exploration of when to replace ReLU with an approximation seems somewhat expectable despite the novelty of the exact approach.

Minor issues:
- 1: "huge" (unscholarly language)
- throughout: "Delphi Mishra et al." - maybe "Delphi by Mishra et al."?
- References: curly brackets in several references such as {USENIX}

Conclusion:
I recommend acceptance given the novelty.

---

> ### Author Response · Authors · 2020-11-15
> **Reply to Reviewer 3**
>
> #### We would like to thank reviewer 3 for the thoughtful comments and efforts towards improving our manuscript.
>
> ##### Question 1. Deriving public meta-parameters from data might impact privacy.
>
> Answer 1:  We sincerely appreciate this instructive review. We have added the following description into the manuscript (page 6):
> Our BTPBT takes the all-$ReLU$ neural network model $M_T$ and training data $D$ as inputs in the server side, and outputs a well-trained polynomial-approximated neural network $P_T$. The $P_T$ is used to provide a privacy-preserving machine learning inference service that takes the client’s encrypted sensitive data $D_{inf}$ as input, and outputs the encrypted prediction result to the client. So our BTPBT does NOT impact the privacy of user's data $D_{inf}$. Our BTBPT shares the same threat model with our baseline Delphi [1].
>
> [1] Mishra, Pratyush, Ryan, Lehmkuhl, Akshayaram, Srinivasan, Wenting, Zheng, and Raluca Ada, Popa. "DELPHI: A cryptographic inference service for neural networks." . In USENIX Security.2020.
>
>
> ##### Question 2. The further exploration of when to replace ReLU with polynomial seems somewhat expectable despite the novelty of the approach.
>
> Answer 2:
> - It is somewhat expectable that we can replace some ReLU without a large accuracy decrease. However, it is unexpectable to decide which channels should be replaced since the accuracy is sensitive to the activation channel [2] and channels are of joint impact on the accuracy. It is also unexpectable to decide which polynomial is better to approximate the ReLU. It seems that higher polynomial approximation has less approximation error. However, the high-degree polynomial approximation (degree>3) will introduce poor training accuracy, due to the exploding gradient of high-degree polynomial.  And previous works usually use degree-2 square polynomial. As far as the author knows, we are the first to support the flexibility in the degree of polynomial which enables a 1.8%-4% accuracy improvement under the same latency constraint.
> - Secondly, as the reviewer 4 mentioned, we have mainly three novel contributions: 1. Channel-wise approximation of activations.  2. More flexibility in the form of the polynomial approximation.  3. BTPBT hyper-parameter search.
>
> [2] He, Yihui, Ji, Lin, Zhijian, Liu, Hanrui, Wang, Li-Jia, Li, and Song, Han. "Amc: Automl for model compression and acceleration on mobile devices." . In Proceedings of the European Conference on Computer Vision (ECCV) (pp. 784–800).2018.
>
> ##### Question 3. Word selection and reading-friendly citation.
>
> Answer 3: We really appreciate reviewer 3 for the thoughtful comments towards improving our manuscript. We have fixed them in the revised manuscript.

---

### Official Review · AnonReviewer4 · 2020-10-29
**Interesting technique for model optimization targeting secure neural network inference**

**Rating:** 7
**Confidence:** 4

**Review:**

Summary:

The main contribution of this paper is a new heuristic for identifying "less useful" activation channels. The authors then propose using simple approximations for activation functions for these channels without compromising network accuracy. The main novelty in the approximation used by the authors is flexibility in the degree of the polynomial approximation. Additionally, the authors propose a new hyper-parameter search strategy (BTPBT) to efficiently search the hyper-parameter space for the optimal approximation parameters.

Score Rationale:
- The paper has three novel contributions:
  - Channel-wise approximation of activations
  - More flexibility in the form of the polynomial approximation
  - BTPBT hyper-parameter search
- As a result of these contributions the authors demonstrate a 40% reduction in total inference latency compared to state of art

Strengths:
- The authors present a thorough description of the experimental procedure that is sufficient to replicate their work
- An informative ablation study is presented to summarize the relative contribution of the various techniques proposed
- The baselines chosen are competitive and allow for a fair evaluation

Weaknesses:
- The main weakness of this work stems from the modest improvement in latency afforded by these techniques
- In particular while secure inference is still orders of magnitude less efficient compared to the corresponding plaintext computation these techniques can only account for a factor of 2 compared with a baseline that implements no secure inference specific network tuning (e.g. Gazelle)

Additional Comments/Questions:
- Is the accuracy of the three networks from Fig 4b comparable?
- Fig. 5 caption has a typo. The (a) and (b) sub-captions should indicate the network name
- What is the source of the ~10% improvement in the runtime of the linear layers when compared with Delphi?

---

> ### Author Response · Authors · 2020-11-15
> **Reply to Reviewer 4**
>
> #### We would like to thank reviewer 4 for the insightful comments and efforts towards improving our manuscript.
>
>
>
> ##### Question 1. Modest improvement (only 2$\times$ improvement) in latency.
> Answer 1:
> -	First, we added new results in the table 2~4 of the revised manuscript to show that SAFENet also has the ability to improve accuracy other than reducing latency. SAFENet is able to reduce latency by 38\% ~  61\% or imporve accuracy by 1.8\% ~ 4\% over prior techniques.
> -	Secondly, SAFENet reduces the total inference time of VGG-16 from 172 seconds to the 104 seconds and improves inference accuracy by more than 0.8% at the same time. We believe that the reduction of 68 seconds inference latency is important for the real-life applications. When we use the Delphi-Fast to approximate more activations than Delphi, shown in Table 4 in the revision, the accuracy of Delphi-Fast is decreased decreased 3.2\% with only a  ~6 seconds latency reduction.  Our work SAFENet improves 4\% inference accuracy and reduces 37.3\% latency over Delphi-Fast at the same time.
> -	Thirdly, SAFENet has a large performance improvement on online time. For example, the online time of our baseline Delphi using CNN-7 on CIFAR-10  is 2.9$\times$ of SAFENet as Table 2 shows. It is a fact that online time is much more important than total running time since the offline phase of total running time can be processed in advance.
>
>
> ##### Question 2. Is the accuracy of the three networks from Figure 4b comparable?
> Answer 2: Yes. The three networks including DRL, PBT and our BTPBT are implemented under the same accuracy threshold, called $A_t$. In the Figure 4b, the accuracy constraint is 85%.
>
>
> ##### Question 3. Fig.5 caption typos.
> Answer 3: We sincerely appreciate this instructive review. We have fixed it in the revised manuscript.
>
>
> ##### Question 4. Source of the 10\% improvement in the runtime of the linear layers when compared with Delphi.
> Answer 4: The source of the 10\% improvement is degree-0 activation polynomial approximation that is a channel pruning. In other words, the convolution in the channel of degree-0 activation is not required to computed. So that HE-based linear operations number is reduced.

---

### Official Review · AnonReviewer1 · 2020-10-29
**Reviews of Reviewer 1**

**Rating:** 6
**Confidence:** 3

**Review:**

The paper proposes an interesting idea where they seek to distinguish between activation channels that are crucial for preserving information flow and approximating the rest with low degree polynomials. The authors show that this doesn't result in decrease in accuracy but leads to speedup in performance.

I think it is a neat trick to exploit the high-dimensional intermediate feature space given that the network only uses a low dimensional subspace of it. It is an interesting use of the idea which has been demonstrated in prior works - low dimensionality of the feature space or that the learning task can be done by randomly switching of a huge fraction of available channels.

However, I think the paper has two main issues which prevents me from providing a more favourable decision.


* Novelty and Comparison
   * The key novelty of this paper is to figure out which channels can be approximated with polynomials and which needs to be retained in its original capacity. This approximation leads to a speedup. While it is a neat modification, I think it is very incremental in its novelty and the performance is not very large either.  The time for one ResNet32 inference on CIFAR100 with SafeNet is 0.62x of Delphi, which is not a very large improvement.
   * It would also be helpful to compare with techniques other than gazelle and Delphi that modifies the neural networks prior to training (eg. binarizing the network, ternarizing the network, fixed point precision etc) so that the accuracy-latency of the trade-off can be put into perspective and compared easily.  When considering methods to speedup Encrypted Prediction as a Service, one should naturally talk about fixed precision networks and binary neural networks (BNNs). In the past, BNNs have shown remarkable speedups in computation without a major hit to accuracy (Sanyal et. al. 2018), which is in fact a major point of this paper. Similarly, the paper should also discuss and compare with techniques that uses modification of encryption schemes specifically meant to optimize activations in NNs (eg. Lou et. al. 2019)

There has been a large body of work in the past few works that claim to improve latency of encrypted prediction. It is getting hard to say whether a technique really provides a speedup unless a more comprehensive experimental survey is done with a series of papers using benchmark datasets and networks. i would encourage the authors to have more techniques in their experimental comparison section.

* Clarity -  I have found the paper very hard to read in general and some major gaps when considering baselines and background discussion.
    * I think the paper would be much easier to read if the citations were modified with something akin to \citep and \citet.
    *  The figures (Fig 1, Fig 3, Fig 4) are very far from being self-contained. I think it is okay to omit some details from the figures But there should be proper explanations in the captions. It is very hard to understand to understand what the notations mean in Fig 1A and what 0L, 4L, and the percentages mean. Similarly Fig 4a, is very congested with overlaps between arrows and figures and it is very hard to read and understand what the different notations mean.

I think, overall, the paper needs an overhaul to first make sure that the various components used in the pipeline are explained properly, make sure the figures are clear, the notations are clear especially in the earlier sections. I think further advantages can be gaining incites from papers that talk about importance of various layers (Zhang et. al. 2019), using low rank layers or representations etc and doing comparisons with other works that claim to provide an improvement in encrypted prediction latency.

Zhang, C., Bengio, S., & Singer, Y. (2019). Are all layers created equal?. arXiv preprint arXiv:1902.01996.

Lou, Qian, and Lei Jiang. "SHE: A Fast and Accurate Deep Neural Network for Encrypted Data." Advances in Neural Information Processing Systems. 2019.

Sanyal, A., Kusner, M. J., Gascon, A., & Kanade, V. (2018). Tapas: Tricks to accelerate (encrypted) prediction as a service.  International Conference in Machine Learning, 2018.

---

> ### Author Response · Authors · 2020-11-15
> **Reply to Reviewer 1**
>
> #### We would like to thank reviewer 1 for the thoughtful comments and efforts towards improving our manuscript.
>
> ##### Question 1: Novelty of channels approximation may be incremental.
> Answer 1: SAFENet has three novel contributions:
> - 1. Channel-wise approximation of activations.  We identify the performance bottleneck of existing cryptographic inference is still activation function. We then propose channel-wise activation approximation to replace more expensive GC-based activation with the cheap BT-based polynomials.
> - 2. More flexibility in the form of the polynomial approximation.  What polynomial should be used to approximate activation? It seems that higher polynomial approximation since it has less approximation error. However, the high-degree polynomial approximation (degree>3) will introduce poor training accuracy, due to the exploding gradients.  And previous works usually use degree-2 square polynomial. As far as the author knows, we are the first to support the flexibility in the degree of polynomial, e.g. we support the hybrid using of degree-3, degree-2 and degree-0 polynomials. Degree-0 polynomials can be seen as channel pruning.
> - 3. BTPBT hyper-parameter search. How to perform channel-wise activation replacement and flexible polynomial at the same time? We model it as a hyper-parameter optimization problem.  This hyper-parameter optimization problem is not trivial. Hand-crafted methods, traditional Deep Reinforcement Learning (DRL) and even Population Based Training method are not suitable for this hyper-parameter optimization problem due to its enormous search space. We propose binary-tree based PBT (BTPBT) algorithm to solve this problem.
>
> ##### Question 2. The performance improvement is not very large (e.g. 0.62x improvement on Delphi for ResNet 32 inference).
> Answer 2:
> - First, we added new results in the table 2-4 of revised manuscript to show that SAFENet also has the ability to improve accuracy other than reducing latency. SAFENet is able to reduce latency by 38\% ~ 61\% or imporve accuracy by 1.8% ~ 4% over prior techniques.
> - Secondly, SAFENet reduces the total inference time of ResNet-32 from 204 seconds to the 128 seconds and improves inference accuracy by more than 0.2% at the same time. We believe that 76 seconds inference latency reduction is important for the real-life applications; When we use the Delphi-Fast to approximate more activations than Delphi, shown in Table 4 in the revision,  the accuracy of Delphi-Fast is decreased 1.6\% with only a  ~6 seconds latency reduction.  Our work SAFENet improves 1.8\% inference accuracy and reduces 35.8\% latency over Delphi-Fast at the same time.
> - Thirdly, SAFENet has a large performance improvement on the online phase. For example, the online time of our baseline Delphi using CNN-7 on CIFAR-10  is 2.9x of SAFENet as Table 2 shows. It is a fact that online time is much more important than total running time, since the offline phase of inference can be processed in advance.
>
> ##### Question 3. Comparison with TAPAS[1] and SHE[2].
> Answer 3: We thank the reviewer for pointing these related works. We added the comparison of TAPAS[1] , SHE[2] and our work SAFENet in the related work of the revised manuscript. TAPAS uses  HE-based binary neural networks which adapts $sign()$ function instead of the activation, thereby suffering from more inference accuracy loss ($<99$\% accuracy on MNIST) than approximated ReLU function ($>99$\% accuracy on MNIST). And TAPAS and SHE suffer from long-latency linear operations since they use a HE scheme called TFHE[3] that does not support ciphertext batching operations yet. For example, TAPAS and SHE take $\sim$2 hour and ~10 seconds respectively to perform one single MNIST inference with ~99\% accuracy. In contrast, our work SAFENet and Gazelle using the hybrid of batched HE and MPC are able to achieve $<1$-second latency with $>99$\% accuracy.
>
> [1] Sanyal, Amartya, Matt, Kusner, Adria, Gascon, and Varun, Kanade. "TAPAS: Tricks to Accelerate (encrypted) Prediction As a Service." . In International Conference on Machine Learning (pp. 4490–4499).2018.
>
> [2] Lou, Qian, and Lei, Jiang. "SHE: A Fast and Accurate Deep Neural Network for Encrypted Data." . In Advances in Neural Information Processing Systems (pp. 10035–10043).2019.
>
> [3] Ilaria Chillotti, , Nicolas Gama, Mariya Georgieva, and Malika Izabach\`ene. "TFHE: Fast Fully Homomorphic Encryption Library." (August 2016).
>
> ##### Question 4. \cite to \citet; more explanation on the captions and notations of figures 1 3 4 for clarity.
> Answer 4: We sincerely appreciate reviewer 1’s instructive review. We have fixed it in the revised paper.  We also appreciate it very much if we would have follow-up comments and suggestions.

---

> > ### Comment · AnonReviewer1 · 2020-11-23
> > **Thank you for the revision**
> >
> > Thank you very much for the helpful comment and the **quite significant** update to the paper and incorporating the suggestions in such a short time.
> >
> > The improvements to the caption of Figure 1b and the overall Figure 4a, the citations, and clarifying the exact quantitative improvements throughout the paper helps to highlight the main results of the paper.
> >
> > Given my revised understanding of the contribution and the significant update to the clarity of the paper, I think I am happy to raise my score to reflect this.

---

> > > ### Author Response · Authors · 2020-11-25
> > > **Thanks for your helpful feedback**
> > >
> > > Dear Reviewer 1,
> > >
> > > We really appreciate your feedback and thank you for the increased score.  Your comments have been very important and valuable for us to improve the work!
> > >
> > > Sincerely,
> > >
> > > Authors

---

### Decision · Program_Chairs · 2021-01-07
**Final Decision**

**Decision:**

Accept (Poster)

**Comment:**

The authors did a nice job of responding to the concerns of reviewers during the discussion phase which increased reviewer scores. Because of this I will vote to accept.

The authors should carefully edit the paper for typos, grammatical errors, and style errors. Some examples:
- Abstract: Make this one paragraph without a line break
- End of 1st paragraph in Intro: "So there is an urge" -> "So there is an urgent"
- Start of 3rd paragraph in Intro: "State-of-the-art cryptographic" -> "The state-of-the-art cryptographic"
- Last paragraph of 2.1: "To solve above" -> "To solve the above"
- End of 2.3: "Compared to the light-weight InstaHide and TextHide, MPC and HE are of advantages in the security guarantees so far." -> "Compared to the light-weight methods InstaHide and TextHide, MPC and HE provide much stronger security guarantees."

I also urge the authors to please double check the reviewer comments when preparing a newer version to ensure all concerns are taken into account.

---

> ### Author Response · Authors · 2021-03-14
> **Reply to program chairs**
>
> We would like to thank the program chairs for the thoughtful comments and efforts towards improving our manuscript. We carefully edit our paper to incorporate all the helpful suggestions from reviewers.